# Investigation on the Anisotropic Wetting Properties of Water Droplets on Bio-Inspired Groove Structures Fabricated by 3D Printing and Surface Modifications

**DOI:** 10.3390/biomimetics7040174

**Published:** 2022-10-24

**Authors:** Ngoc Phuong Uyen Mai, Po-Yu Chen

**Affiliations:** Department of Materials Science and Engineering, National Tsing Hua University, Hsinchu 300044, Taiwan

**Keywords:** unidirectional wetting, grafting, atmospheric pressure plasma, surface modification, 3D printing

## Abstract

The self-driving structure to orientate the water movement has attracted considerable attention. Inspired by the wedgelike structures of biological materials in nature, such as spider silks and cactus spines, anisotropic spreading can be realized by combining Laplace pressure gradient and hydrophilic surface. In this study, a series of groove patterns were fabricated by a combination of 3D printing and surface modification. PLA pattern was modified by the atmospheric pressure plasma, followed by grafting with hydrolyzed APTES. This work reports the anisotropic transport of water droplets on a series of designed dart-shaped groove patterns with specific angles in the main arrow and tail regions. This structure can induce capillary force to regulate droplets from the main cone to two wedgelike, whereas the droplets are hindered toward the opposite side is oat the vicinity of the groove’s tail. By means of the experiment, the mechanism of water transport in this pattern was revealed. This study can contribute a potential approach to manipulate and apply anisotropic wetting in many fields.

## 1. Introduction

Anisotropic wetting behavior without supplying external energy is of important interest in many research fields, such as micro-fluidic devices, bio-sensors, water/fog collection, and industrial applications [1,2,3,4,5]. A variety of creatures in nature have the ability to fulfill unidirectional liquid transport. For example, spider silk possesses periodic spindle knots and joints to produce droplet motion that is utilized to capture water from fog effectively [6,7]. The *Nepenthes alata* can make use of the oriented capillarity to transfer liquid up to the peristome and create a slippery trap that catches small insects [8,9]. Taking advantage of slippery surface, liquid-repellent microstructured surface could further investigate in fluid handling, transportation, and optical sensing, which is considered as the nonwetting structures’ counterpart [10,11,12,13,14]. The feature of Gecko toe pads encompasses adhesion and self-cleaning by ubiquitous spatulae, which induces van der Waals forces with surfaces. To imitate the motions of gecko feet, the splayed liquid crystal elastomer (LCE) micropillars have been investigated to synthesize the properties of switchable dry adhesives and dynamic self-cleaning [15,16,17]. In an arid environment, the beetles back in the Namib desert have the capacity of collecting water from fog through the bumpy surface, which constitutes alternating hydrophobic and hydrophilic regions [18]. The advanced bioinspired texture for the water-harvesting system has been uncovered, in which the surface integrates the beetle bump property with an asymmetric slope inspired by cactus spines for capillary-driven and molecular-scale smooth lubricant on nanoscale texture -based a pitcher-plant [19,20].

A unidirectional wetting on a wedgelike groove is typically based on gradient surface tension (pH, temperature, UV light, etc.), Laplace pressure gradient, wettable surface, or an anisotropic structure [21,22,23,24,25]. In previous reports, the preparation of a wettable surface is commonly fabricated by alternating hydrophilic groove and hydrophobic background or thermocapillary migration, which drive liquid from less hydrophilic to more hydrophilic regions or from warm to cold areas, respectively. In addition, an anisotropic structure also plays a role in enhancing unidirectional wetting due to the gradient Laplace pressure, which is built by photolithography, a complicated and costly method [26,27,28]. Consequently, a new means of one-way transport should be surmounted the limitation.

In this study, taking the advantage of wedge-shaped features for anisotropic liquid spreading, we designed a dart-shaped structure, which consists of a main wedgelike at the main cone and two additional wedgelike at the pattern’s tail. The efficiency of directional transport with a continuous water supply would be observed through the features from the function of the transition zone under the given condition. The preparation procedure seemed to be simpler than the aforementioned methods. Instead of heterogeneous wettability, the groove surface was modified uniformly to be hydrophilic, thus liquid was easy to spread over it. The surface profiles and morphologies of the patterns were observed by laser confocal microscopy, atomic forces microscopy, and chemical composition was analyzed by electron spectroscopy. Regarding the dart-shaped groove, the gradient Laplace force at the main wedgelike induces to orientate transport direction, while the geometry of two wedgelike at the neighboring groove’s tail obstructs droplets from moving to the opposite side. We expect that these patterns could lead to the inspiration of unidirectional wetting patterns and potential practical applications.

## 2. Materials and Methods

### 2.1. Dart-Shaped Structure Design

According to the study of the cacti [19], we conceived to build a three-dimensional pattern, which is a concave quadrilateral, with two of its sides moving inward. This structure contains two zones; the main zone is similar to the conical structure, which serves as a driving force, while the two cones at the pattern tail play a role in blocking fluid transport in the opposite direction.

We studied the mechanism of liquid flow in the dart-shaped groove, the five factors that manage directional movement include the length of the groove (l), the thickness of the groove (t), the width of the groove’s tip (w), the tilt angle of the main cone (α), and the tilt angle of two cones at groove tail. The design parameters of this groove can be shown in Figure 1.

### 2.2. Experiments

#### 2.2.1. Preparation of Dart-Shaped Pattern

Instead of using photolithography, we employed fused deposition modeling 3D printing to prepare designed patterns. The Ultimaker S3 (Ultimaker, The Netherlands) was used to print the dart-shaped groove patterns. The layer resolution was 20 μm, and the print precision size was 0.01 mm, leading to a high quality of the printed sample. The benefit of the printer was cost-effectiveness, easy to maneuver, and fabricate an amount of sample at the same time. The patterns were first designed by the Solidworks software SolidWorks 2021 SP2.0 (Dassault systems S.A, Waltham, MA, USA) then imported the stereolithographic file into Ultimaker Cura (Ultimaker, The Netherland), and finally transferred to the printer. It constructs prototypes via melting a solid plastic filament and depositing it onto the platform to form stacked layers. Figure 2a indicates the 3D model input to the printer, and the Ultimaker 3D printer illustration is shown in Figure 2b.

#### 2.2.2. Structural and Morphological Characterization

The quality of the modified surface was evaluated by the atomic force microscopy (Bruker, model: Dimension ICON) under the tapping mode to observe the morphology, the electron spectroscopy for chemical analysis (ESCA, ULVAC-PHI, PHI 5000 Versaprobe II), and the contact angle goniometer (First Ten Angstroms 1000, Portsmouth, NH, USA) for wettability. The measurement of the contact angle took place at 26 °C and 60–69% relative humidity, respectively. We observed the structure of a 3D printed dart-like pattern with a laser scanning confocal microscope (Keyence VK-X, Osaka, Japan). Particularly, the change in the groove thickness has been conducted from 0.2 mm to 1.0 mm. We realized that when the thickness is shallow (less than 0.8 mm), resulting in droplets overflow to the borders and the spreading behavior seems to be disordered. In contrast, if the groove depth is larger than 0.8 mm, the plasma beam cannot activate the surface, leading to grafting ineffectively. In terms of the groove width, in the case of fixed other parameters, the width becomes wider, which causes a smaller Laplace pressure gradient and cannot evaluate the wetting behavior. However, to guarantee the effective unidirectional wetting, all parameters of the groove have to be changed in accordance with the width that would be difficult to design. As a result, three fixed parameters included the total length of a single groove was 6.0 mm, the width of the groove was 1.5 mm, and the thickness of the groove was 0.8 mm. The tilt angle of the main wedgelike was designed to vary from 5° to 25°, and that of two wedgelike at groove tail vary from 20° to 50°.

#### 2.2.3. Surface Modification

The contact angle of untreated polylactic acid (PLA) was measured to be approximately 73°. Hence, the printed patterns were treated with atmospheric pressure plasma at a specific working distance of 5 mm between the surface and the plasma jet to activate the wettable surface by introducing temporary free radicals and reactive species. The samples were followed by immediately placed in a hydrolyzed solution with a concentration of 2% APTES in Ethanol (95%) to achieve stable hydrophilicity. As a result, the contact angle of the modified surface could be reduced to about 42° and its hydrophilicity was improved.

#### 2.2.4. Observation of Wetting Behavior

A digital camera (D750, Nikon, Tokyo, Japan) was used to observe and record the dynamic wetting of water droplets on flat and patterned PLA surfaces. The needle of the contact angle goniometer was placed in the middle of the pattern and continuous droplets were supplied, as illustrated in Figure 2c. The direction and traveling distance of the water flow were evaluated. The videos were recorded at 660 frames per second. Colored deionized water of 500 μL in total volume was used to visualize the wetting behavior onto the untreated and modified patterns at room temperature and relative humidity of 24–26 °C and ~53%, respectively.

## 3. Results

### 3.1. Surface Characterization

#### 3.1.1. Static Contact Angle Measurement

The original surface exhibited a contact angle of 73.35°± 1.65°. After AP plasma treatment, the water contact angle of 2 μL water droplet decreased to ~58°. However, the hydrophilic surface generated from the AP plasma treatment tended to recover to its initial wettability, because free radicals were highly unstable and easy to undergo autoxidation reactions [29]. Consequently, in order to stabilize the hydrophilicity of plasma-treated PLA, surface functionalization was attained by grafting hydrolyzed APTES onto plasma-treated PLA surface at 50 °C for 6 h in a vacuum oven. The plasma-treated, hydrolyzed APTES grafted surface reached a contact angle of 42.42°± 1.11°, as shown in Figure 3a. To substantiate the efficiency of the grafting technique, the long-term durability of hydrophilicity was assessed, as shown in Figure 3b. The hydrophilicity of the surface after grafting was well maintained, with an average water contact angle of

45.8°± 1.5° for 20 days, while the contact angle recovered to 62.6°± 1.3° for samples merely underwent AP plasma treatment.

#### 3.1.2. Chemical Composition Analysis

Figure 4 offers the change in elemental composition by the peak fitted narrow scan C1s, O1s, and N1s spectrums after each treatment. In terms of the C1s peaks of the plasma-treated sample, two peaks appear at 285.75 eV and 287.99 eV, corresponding to the C−O and O−C=O bonds, respectively. The intensity of these peaks increases and then decreases substantially compared to untreated PLA and hydrolyzed APTES-grafted PLA, respectively. C1s spectrum of grafting reveals C−C or C−Si at 283.89 eV with relatively low intensity. The curve fitting of the O1s spectrum for the plasma method shows −OH group on two peaks, HO−C=O bond at 523.45 eV and C−O−C or C−OH bonds at 532.78 eV. Particularly for the grafting surface, the binding energy of O1s appears Si(−O)_2_ at 531.09 eV, and Si(−O)_3_ at 532.76 eV, which proves two or three hydroxyl groups −OH from grafting solution to the reactive plasma-treated PLA. Regarding the N1s spectrum, the presence of −NH_2_ at 298.39 eV and − HNC= O at 400.13 eV is only observed for the grafted surface.

#### 3.1.3. Surface Morphology Characterization

The AFM images exhibit the surface morphology with the grafting process at different reaction durations, as shown in Figure 5. The phase image contains the bright and dark areas representing regions with a low and much high phase shift. In detail, the reaction duration of 2 h could be the irregular APTES molecules covering the surface. For 4 and 7 h of the reaction durations, the defects appear substantially to cause the difference in contrast. Especially, the reaction duration of 6 h displays a slight differential contrast, which indicates the distribution of APTES onto the PLA surface is relatively uniform.

### 3.2. Dynamic Wetting Behavior on 3D-Printed Dart-Shaped Pattern Array

The dart-like structure contains a wedgelike groove at the main zone and another two smaller ones at the second zone along the symmetric axis. Combining 3D printing and surface modification, this model is fabricated for further investigation. To optimize the design parameters, we set up and evaluate two cases, simultaneously. For the first case, the β angle is fixed at 20°, while α angles vary from 5°, 10°, to 20°, as shown in Figure 6a. For the second case, the α angle is fixed at 20°  and the β angles vary from 20°, 30°, 40°, to 50°, as shown in Figure 6b. Other parameters are fixed with specific values for all samples (in Section 2.2.2). Based on the laser scanning confocal microscope measurement, the accuracy of the printed samples can reach 99% in length and 98% in thickness for all actual patterns compared to the designed patterns.

#### 3.2.1. Dynamic Wetting Behavior of Case 1

The droplet starts off in the middle groove of the sample and we observe the spreading phenomena through the maximum distance that water droplets can eventually reach (with 18 mm on each side). A series of images show the directional movement on the dart-shaped patterns of untreated and modified surfaces.

As shown in Figure 7, the untreated pattern for an α angle of 5° illustrates the water spreading in a short distance of ~2.6 mm toward the predominant direction; the resistance on the surface hinders the droplet from flowing. During the same period of time for the modified one, droplets move simultaneously in both directions within 5 s, then reach 18 mm toward the predominant side and 11 mm toward the opposite side, respectively. Similarly, for sample with α angle of 10°, the wetting distance of ~7.7 mm is observed within 20 s on the untreated pattern. Droplets easily spread both sides on the modified surface with 18 mm and ~9.2 mm to the predominant and opposite sides, respectively, as shown in Figure 8. Thus, α = 5° and 10° samples in the main cone are not capable of unidirectional wetting, owing to the area proportion of two wedge edges at the tail’s pattern seems to be insignificant as opposed to the total area, which is analogous to groove channel. Particularly, when the α angle increases to 25°, the anisotropic wetting is realized with the highest ratio of 6.8. In detail, droplets wet on the modified pattern with 18 mm from the tip to the tail of each groove and merely spread ~2.6 mm toward the opposite side, where droplets are impeded by the neighboring two wedge edges, as shown in Figure 9. These results are proved by the ratios presenting the traveling distance of the predominant side to the opposite side (Figure 10), which are 1.6, 1.9, and 6.8 for 5°, 10°, and 25° in α angles, respectively.

#### 3.2.2. Dynamic Wetting Behavior of Case 2

In terms of varying β angles, droplets only spread 1.6 mm on the untreated pattern with 20° β angle. Within 15 s, the modified one exhibits effective single-way transport of 18 mm in length and spread only ~2.9 mm toward the opposite side, as shown in Figure 11a,c. The phenomenon on the modified sample of 30° in β angle is observed analogously to that of 20° sample. A short wetting distance of 2.4 mm is observed on the opposite side, then droplets commence to wet 18 mm toward the principal direction within 30 s. Water droplets only travel 6 mm on the untreated pattern within 32 s, as shown in Figure 11b,c. For β angle of 40°, droplets spread 2.8 mm on the nonadherent surface, as shown in Figure 12. In contrast, the modified pattern still effectively regulates unidirectional wetting with 18 mm within 21 s, it extends a short distance of ~2.2 mm toward the opposite direction in [3]. The dynamic wetting behavior of water droplets in the 3D-printed dart-shaped pattern with β angle of 50° is shown in Figure 13. In Figure 14, it is conspicuous to attain high ratios (6.3–8.3). For β angle of 50°, anisotropic spreading is not realized, droplets break the opposite side with a length of 7.2 mm, leading to a small ratio of 2.5. This may be caused by the two wedge edges being over an angle threshold and becoming a single wedgelike structure. Compared to the modified one, the untreated surface has a short transport distance of 6 mm within 35 s (Figure 13a).

In summary, two major findings are the wetting performance on the modified sample is better than that of the untreated one, and the criteria of dart-shaped groove for unidirectional spreading include α angles between 20–25° and β angles in the range of 20–40°. The ratios of predominant wetting distance over opposite wetting distance for these samples are higher than 6.0. To elucidate the aforementioned summary, the wetting mechanism of this model is proposed.

#### 3.2.3. Wetting Behavior of Dart-Shaped Groove

The anisotropic wetting mechanisms owing to hydrophilicity surface and asymmetric structure are elucidated and explained. For instance, Geng et al. [30] integrated superhydrophilic surfaces with two-dimensional asymmetrical hydrophobic barriers to fulfill unidirectional wetting. In this study, the Laplace pressure gradient leads to shape change of droplets, which is explained for the dart-like structure, as illustrated in Figure 15. For the convex meniscus in the opposite flow direction, the atmospheric pressure (p_a_) is smaller than the inward pressure of droplets (p_w_), which increases the attractive force within the liquid (cohesion force) and reduces the interaction with the wall. Hence, the liquid droplet is pinned at this position. For the predominant flow direction, the droplet with concave surfaces at both edges stimulates the capillary flow and allows droplets to stick on the groove wall (adhesion force) [31]. Let p_w1_ and p_w2_ denote droplet pressures at edge 1 and edge 2. When p_w1_ is larger than p_w2_, the droplet would move from the main wedgelike structure to the tail, in agreement with experimental observation.

Generally, the mechanism is divided into five stages to describe how the dart-shaped structure fulfills unidirectional wetting, as shown in Figure 16. The essential factors contribute to orientating the flow, including hydrophilic surface and capillary pressure (Laplace pressure) generated by the wedgelike structure, which depends on the adhesion cohesion and surface tension.

Initially, droplets are dripped in the middle region of the sample, then fill entire grooves. The yellow arrows point in the direction of the droplet’s movement. Droplets continue wetting onto the wall of the groove to cause a force on the liquid at the edges that propel the droplet from moving along the longitudinal channel. Following the upward direction of the second stage, the end of the groove channel is the main wedge edge, which means Laplace pressure gradient force is produced to attract water droplets. In the downward direction, after escaping the channel, droplets are hindered by the geometry of the vicinity of two wedgelike from spreading and touching the groove wall. Hence, the blocking ability of the dart-shaped structure is one of the main keys to regulating the anisotropic wetting. For the third and fourth stages in the upward direction, droplets keep filling the groove’s tail, followed by Laplace pressure-driven wetting. Reaching the next groove, the process is repeated until approaching the end side of the pattern. Taking the successful cases into account, it demonstrates that if the two wedge edges are getting sharper and the proportion of their area is proper, it becomes a barrier to obstruct droplets from flowing toward the opposite side; and thus, it enhances unidirectional wetting.

## 4. Conclusions

The cooperative effect of uniform hydrophilicity and Laplace pressure gradient enhances the unidirectional wetting of water droplets without an external energy source. In this study, we designed and fabricated a series of dart-shaped grooves by 3D printing, then proceeded with the surface modification to optimize the anisotropic spreading capacity. The major findings of this research are concluded as follows. First, the 3D printed dart-like structure achieved a high accuracy compared to the original designs. Secondly, the effectiveness of the wettable surface was verified by the observation of the long-term durability and surface analyses (AFM, ESCA, LCSM). Finally, the dart-shaped structure with optimized structural parameters was clarified to enhance unidirectional transport following from the main wedgelike to the tail of each groove and the wetting mechanisms were proposed.

## Figures and Tables

**Figure 1 biomimetics-07-00174-f001:**
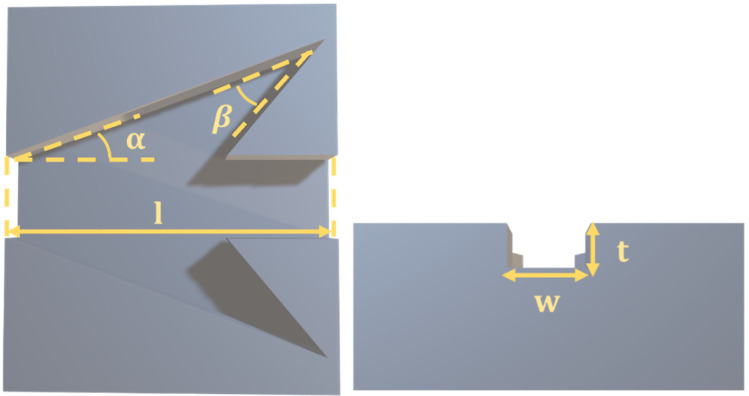
The design parameters of a 3D dart-shaped model.

**Figure 2 biomimetics-07-00174-f002:**
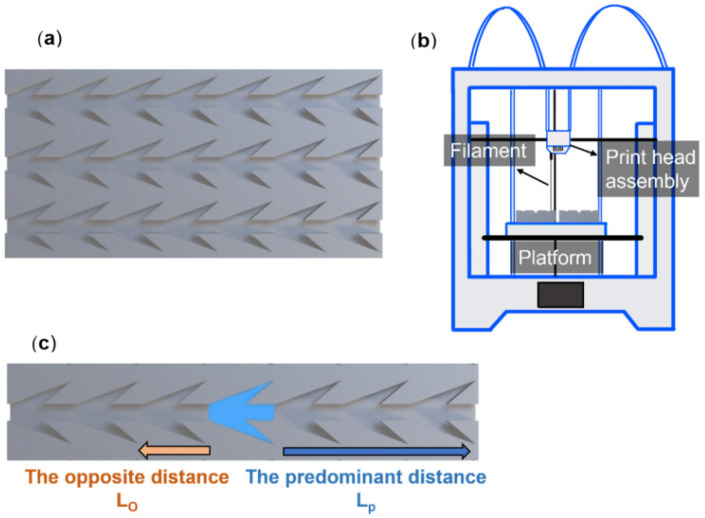
(**a**) Three-dimensional model of dart-shaped structure array; (**b**) The illustration of FDM 3D printer; (**c**) The length of directional wetting.

**Figure 3 biomimetics-07-00174-f003:**
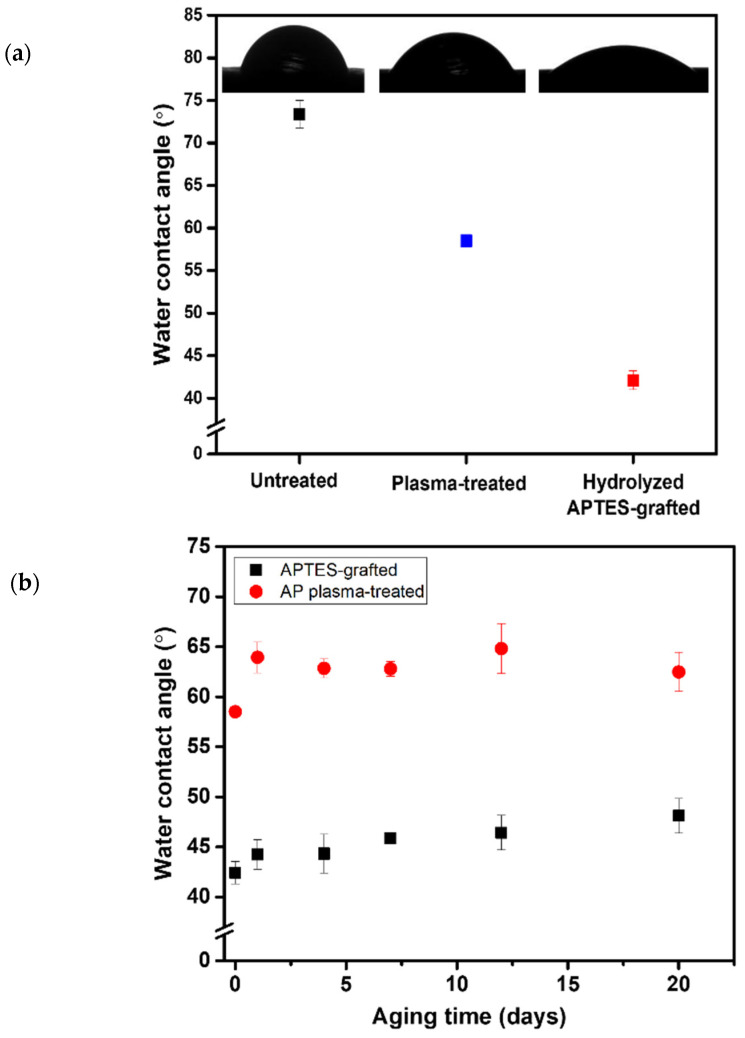
(**a**) Plot of water contact angle versus and the CCD images of contact angle correspond with untreated, plasma-treated, and hydrolyzed APTES-grafted; (**b**) Plot of water contact angle versus aging time.

**Figure 4 biomimetics-07-00174-f004:**
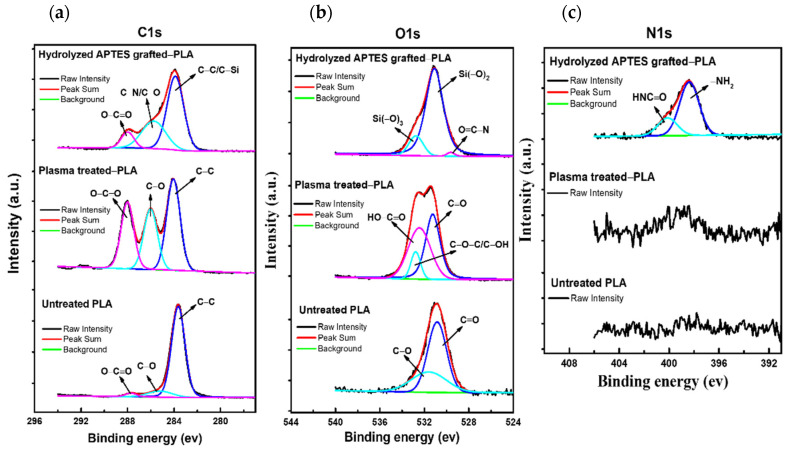
The comparison of the ESCA spectrum for each treatment includes: (**a**) C1s spectrum; (**b**) O1s spectrum; (**c**) N1s spectrum.

**Figure 5 biomimetics-07-00174-f005:**
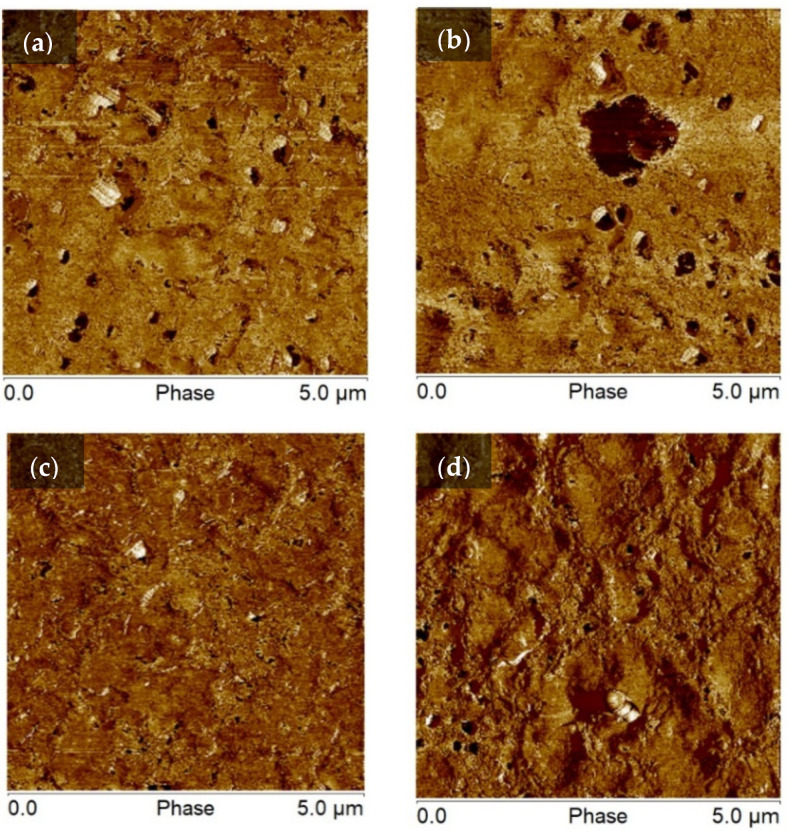
Phase AFM images of the grafting process with the reaction time of (**a**) 2 h; (**b**) 4 h; (**c**) 6 h; (**d**) 7 h.

**Figure 6 biomimetics-07-00174-f006:**
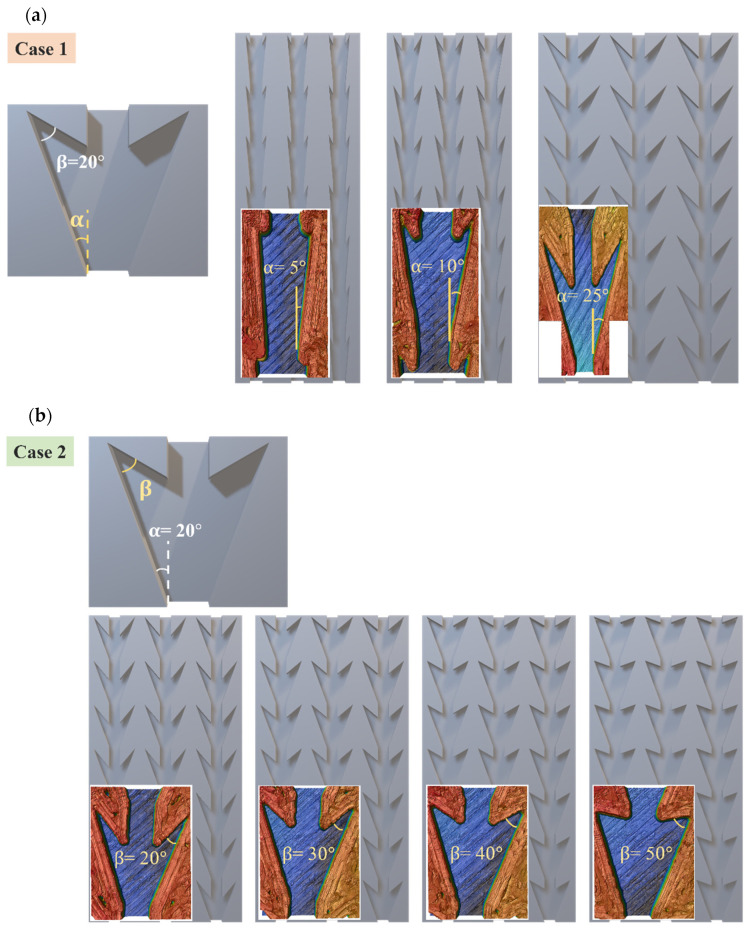
The patterns with different (**a**) α angles; (**b**) β angles and the laser confocal images (color images) show the structural contour of the design structure (gray image). The pattern is 42 mm in total length.

**Figure 7 biomimetics-07-00174-f007:**
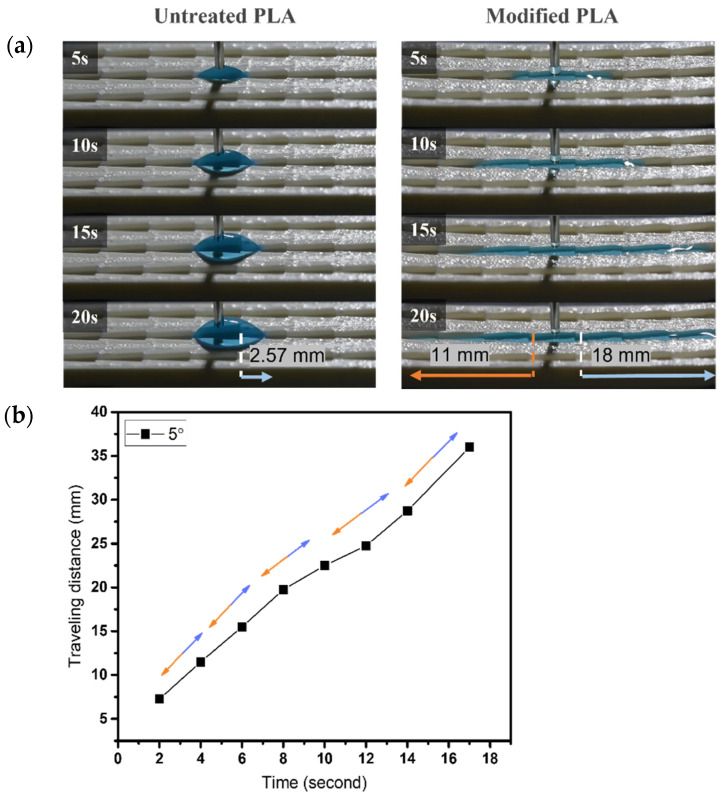
(**a**) The dynamic wetting behavior of water droplets in the 3D-printed dart-shaped pattern with α angle of 5°; (**b**) Plot of traveling distance versus time with the directions of droplet movement wetting of modified pattern. The orange arrow denotes the opposite side, and the blue one denotes the predominant side.

**Figure 8 biomimetics-07-00174-f008:**
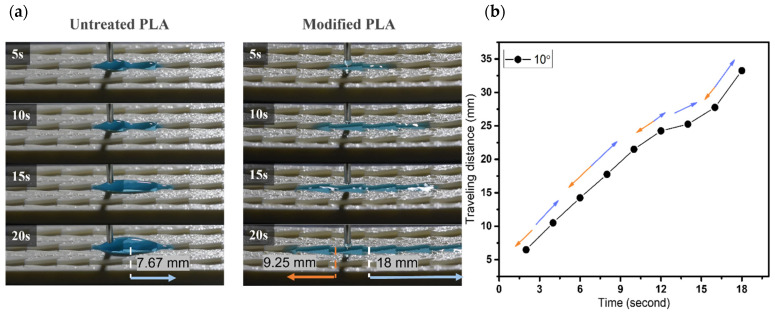
(**a**) The dynamic wetting behavior of water droplets in the 3D-printed dart-shaped pattern with α angle of 10°; (**b**) Plot of traveling distance versus time with the directions of droplet movement wetting of modified pattern. The orange arrow denotes the opposite side, and the blue one denotes the predominant side.

**Figure 9 biomimetics-07-00174-f009:**
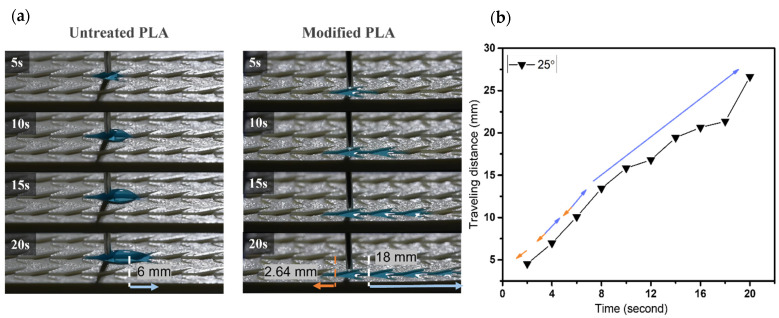
(**a**) The dynamic wetting behavior of water droplets in the 3D-printed dart-shaped pattern with α angle of 25°; (**b**) Plot of traveling distance versus time with the directions of droplet movement wetting of modified pattern. The orange arrow denotes the opposite side, and the blue one denotes the predominant side.

**Figure 10 biomimetics-07-00174-f010:**
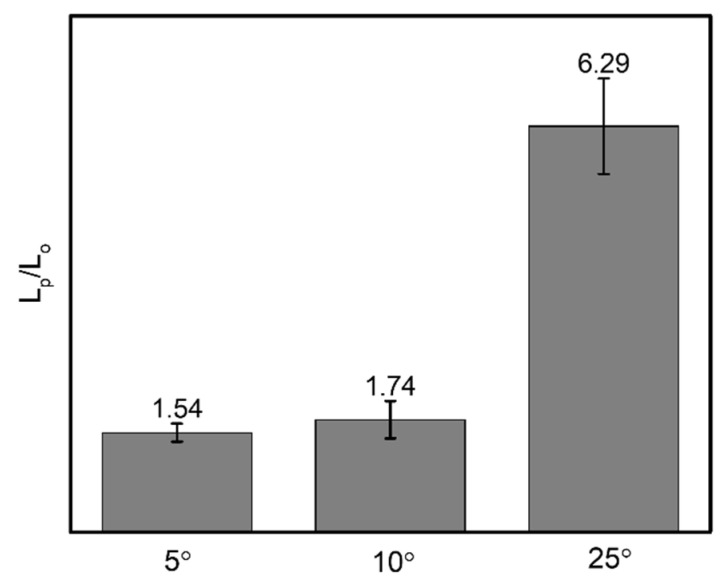
The ratio presents the predominant wetting distance (L_p_) to the opposite one (L_o_) corresponding to various α angles.

**Figure 11 biomimetics-07-00174-f011:**
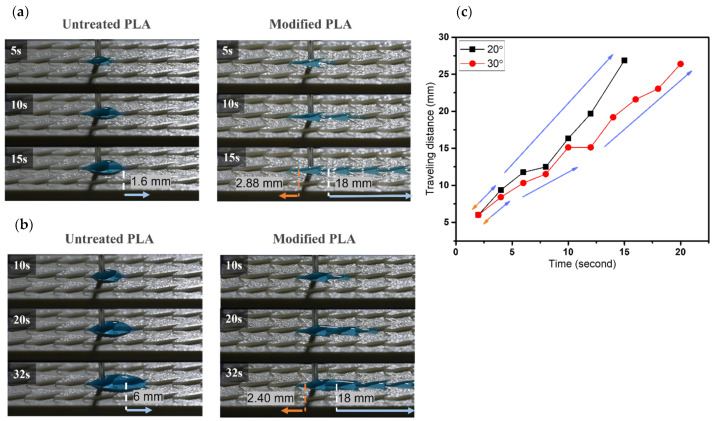
The dynamic wetting behavior of water droplets in the 3D-printed dart-shaped pattern with β angle of (**a**) 20°; (**b**) 30°; and (**c**) Plot of traveling distance versus time with the directions of droplet movement wetting of modified pattern. The orange arrow denotes the opposite side, and the blue one denotes the predominant side.

**Figure 12 biomimetics-07-00174-f012:**
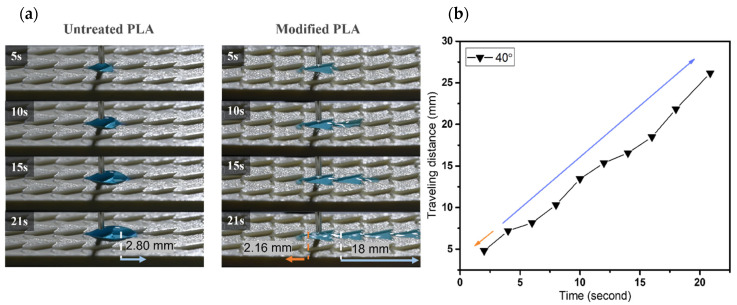
(**a**) The dynamic wetting behavior of water droplets in the 3D-printed dart-shaped pattern with β angle of 40°; (**b**) Plot of traveling distance versus time with the directions of droplet movement wetting of modified pattern. The orange arrow denotes the opposite side, and the blue one denotes the predominant side.

**Figure 13 biomimetics-07-00174-f013:**
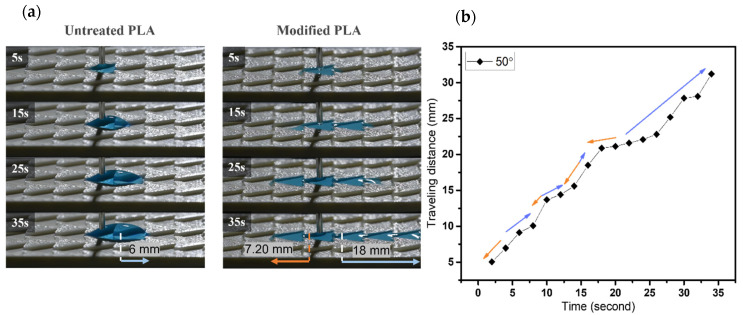
(**a**) The dynamic wetting behavior of water droplets in the 3D-printed dart-shaped pattern with β angle of 50°; (**b**) Plot of traveling distance versus time with the directions of droplet movement wetting of modified pattern. The orange arrow denotes the opposite side, and the blue one denotes the predominant side.

**Figure 14 biomimetics-07-00174-f014:**
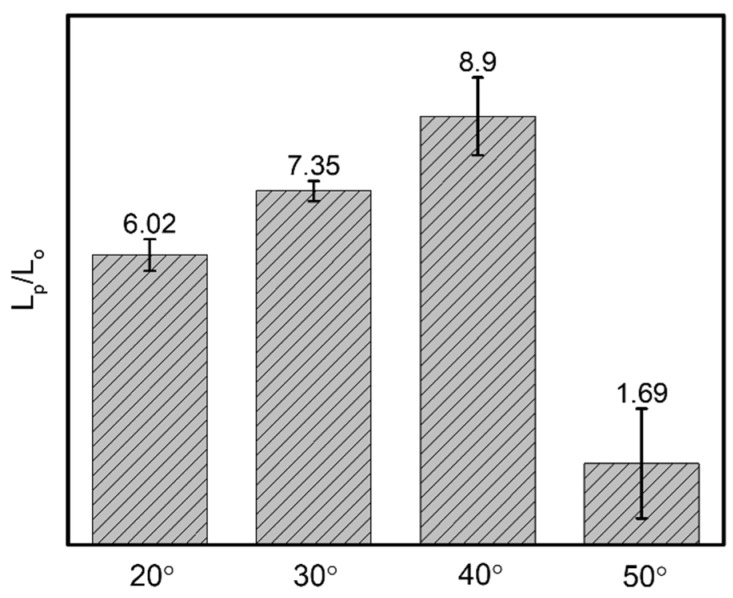
The ratio presents the predominant wetting distance (L_p_) to the opposite one (L_o_) corresponding to various β angles.

**Figure 15 biomimetics-07-00174-f015:**
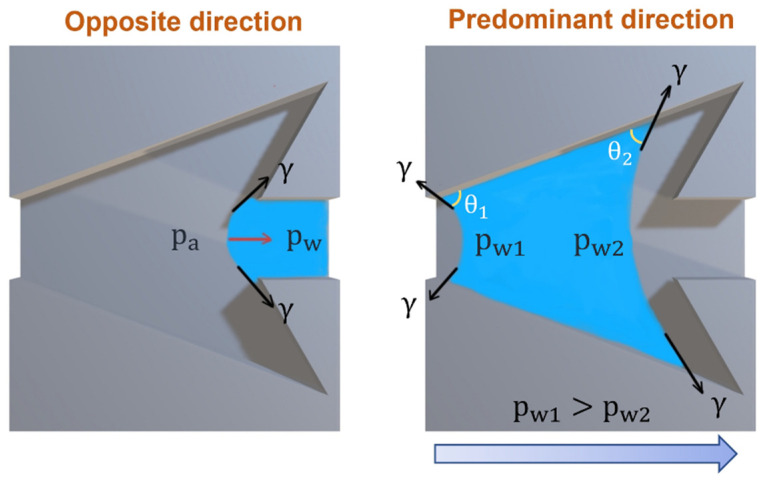
The illustration presents the change in droplet shape with differential pressure.

**Figure 16 biomimetics-07-00174-f016:**
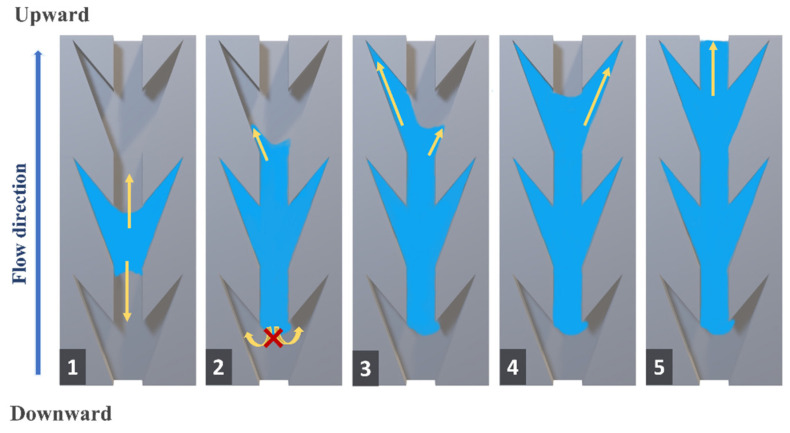
The illustration exhibits wetting mechanisms of dart-shaped structures for unidirectional wetting behavior.

## Data Availability

Not applicable.

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
