# Peer review of "Investigation on the Anisotropic Wetting Properties of Water Droplets on Bio-Inspired Groove Structures Fabricated by 3D Printing and Surface Modifications"

_biomimetics, 2022, doi:10.3390/biomimetics7040174_

Round 1

Reviewer 1 Report

In this review, the authors combined asymmetric wedge structures with hydrophilic surfaces by imitating the spines of a cactus and designed a series of hydrophilic grooves that can achieve asymmetric spreading of liquids. This article contains the analysis of various cases of asymmetric spreading of liquids in asymmetric grooves. The asymmetric spread behavior of the liquid is rationalized by dividing it into five steps, which is reasonable and explicit. This novel work fits well with the journal Biomimetics and it will be inspiring to researchers working on microfluidic and fluid manipulation surfaces. The following comments are recommended to be addressed before publication.

1. Compared to photolithography, the accuracy of fused deposition 3D printers is reduced, so why would this technology be used, is it because PLA surface is easy to adhere to the coating?

2. If the printing accuracy of the 3D model is not ideal, is the tip of the two wedge junctions in the groove sharp, will it have an impact on the asymmetric spreading of the fluid?

3. The authors have discussed the effect of various angles of the wedge structure on fluid diffusion; would a change in the depth and width of the grooves have an effect on these results?

4. Could the authors analyze the mechanism of asymmetric spreading of liquids in more details, for example, drawing the relationship between pressure and droplet shape and with droplet asymmetric spreading distance in the figure? Some articles similar to this one can be introduced, such as: Materials Horizons, 2018, 5(2): 303-308

Reviewer 2 Report

In this work, Mai and Chen reported the manipulation of liquid transport on topographical surfaces. Specifically, 3D printing technique was applied to fabricate a series of groove patterns to direct the water droplet sliding on the surface. Overall, this paper is interesting and reports significant contribution to this bio-inspired surface research field. I support its publication on Biomimetrics after the following key comments are addressed.

1 Biomimetic surfaces are a very popular research topics and there are many efforts to manipulate interfacial water droplet transport. This work only has 18 references, which are not sufficient to knowledge previous efforts in this field. The authors should highlight those works (pillar surfaces, liquid-infused porous surfaces) in this field, e.g., Adv. Mater. 2015, 27, 6828; Nature 2016, 531, 78; Nature 2011, 477, 443; Nature 2018, 559, 77; Sci. Adv. 2021, 7, abi7607; Adv. Mater. 2022, 34, 2108788.

2 The authors should report contact angle hysteresis to provide underlying mechanisms behind the unidirectional liquid transport.

3 What is the effect of the width of the groove structure on the dynamics of anisotropic wetting?
